# Elucidating the Genetic Relationships on the Original Old Sicilian *Triticum* Spp. Collection by SNP Genotyping

**DOI:** 10.3390/ijms232113378

**Published:** 2022-11-02

**Authors:** Maria Carola Fiore, Sebastiano Blangiforti, Giovanni Preiti, Alfio Spina, Sara Bosi, Ilaria Marotti, Antonio Mauceri, Guglielmo Puccio, Francesco Sunseri, Francesco Mercati

**Affiliations:** 1Council for Agricultural Research and Economics (CREA), Research Centre for Plant Protection and Certification, 90011 Bagheria, Italy; 2Stazione Consorziale Sperimentale di Granicoltura per la Sicilia, Santo Pietro, 95041 Caltagirone, Italy; 3Department AGRARIA, University Mediterranea of Reggio Calabria, 89122 Reggio Calabria, Italy; 4Council for Agricultural Research and Economics (CREA), Research Centre for Cereal and Industrial Crops, 190, 95024 Acireale, Italy; 5Department of Agricultural and Food Science, Alma Mater Studiorum, University of Bologna, Viale Fanin, 40127 Bologna, Italy; 6National Research Council (CNR) of Italy, Institute of Biosciences and Bioresources (IBBR), 90129 Palermo, Italy

**Keywords:** biodiversity, bio-morphological traits, single-nucleotide polymorphism (SNP), *Triticum*, wheat landraces

## Abstract

Several *Triticum* species spread in cultivation in Sicily and neighboring regions over the centuries, which led to the establishment of a large genetic diversity. Many ancient varieties were widely cultivated until the beginning of the last century before being replaced by modern varieties. Recently, they have been reintroduced in cultivation in Sicily. Here, the genetic diversity of 115 and 11 accessions from Sicily and Calabria, respectively, belonging to *Triticum* species was evaluated using a high-density SNP array. Einkorn, emmer, and spelta wheat genotypes were used as outgroups for species and subspecies; five modern varieties of durum and bread wheat were used as references. A principal coordinates analysis (PCoA) and an unweighted pair group method with arithmetic mean (UPGMA) showed four distinct groups among *Triticum* species and *T. turgidum* subspecies. The population structure analysis distinguished five gene pools, among which three appeared private to the T. aestivum, *T. turgidum* subsp. *Turgidum,* and ‘Timilia’ group. The principal component analysis (PCA) displayed a bio-morphological trait relationship of a subset (110) of ancient wheat varieties and their wide variability within the *T. turgidum* subsp. *durum* subgroups. A discriminant analysis of principal components (DAPC) and phylogenetic analyses applied to the four durum wheat subgroups revealed that the improved varieties harbored a different gene pool compared to the most ancient varieties. The ‘Russello’ and ‘Russello Ibleo’ groups were distinguished; both displayed higher genetic variability compared to the ‘Timilia’ group accessions. This research represents a comprehensive approach to fingerprinting the old wheat Sicilian germplasm, which is useful in avoiding commercial fraud and sustaining the cultivation of landraces and ancient varieties.

## 1. Introduction

Wheat is one of the key crops to ensure that food and nutritional needs are provided for an ever-growing human population in the future. This is attributable to its ability to adapt to a wide range of climates worldwide. Wheat and its derivatives are foreseen to provide about 20% per capita consumption of protein and calories for an estimated global population of 9.7 billion people in 2050 [1] by contributing to essential components to the human diet [2].

Wheat represents the second most important cereal grain after maize and is the most widely grown crop in the world, with a cultivated area of approximately 219 million hectares and a production of about 760 million tons [3]. Bread or common wheat (*Triticum aestivum* L.) comprises roughly 90% to 95% of the total wheat production worldwide, whereas durum wheat (*Triticum turgidum* L. subsp. *durum* (Desf.) Husn.) represents just 5–7%. However, durum wheat is a staple food in Mediterranean countries due to its adaptation to hot dry conditions and is useful for producing pasta, couscous, bulgur, and bread [4]. Italy is the country leader in EU production and is the second-largest world producer after Canada with a production of 3,885,216 tons [5], a leading role attributable to the economic importance of the pasta industry.

Wheat belongs to the grass family *Poaceae* and the tribe *Triticeae*. Einkorn wheat (*T. monococcum* ssp. *monococcum,* 2n = 2x = 14, AA) is the earliest diploid species of cultivated wheat. Tetraploid wheat (2n = 4x = 28, AABB) includes emmer (*T. turgidum* ssp. *dicoccum*), which was widely cultivated before bread wheat [6], and durum wheat (*T. turgidum* ssp. *durum*), which was cultivated mainly in the Mediterranean basin [7]. Bread wheat *(T. aestivum* L.) is an allohexaploid (2n = 6x = 42, AABBDD) that arose from the interspecific hybridization of *T. turgidum* ssp. *dicoccum* (AABB)) and *Aegilops tauschii* Coss. (DD [8].

The most significant effect of domestication first and breeding later has been a reduction in the genetic diversity of cultivated wheat species [9,10]. In Italy, the genetic improvement of durum wheat was launched in the early 1900s with the exploitation of the genetic variability available in Southern Italy. Moreover, the introduction of exotic landraces from North Africa and West Asia [11] also contributed to the maintenance of a broad genetic variability within Sicilian ecotypes.

Despite the improvements in yield and end-use quality [12,13], modern durum wheat varieties require higher chemical inputs that have a negative impact on the environment, which determined a genetic variability loss due to the replacement of wheat landraces [14,15].

Climate change impacts crop production worldwide, which now requires a large germplasm collection as a source of new traits, especially for bread and durum wheat, which represent the staple food for about 40% of the world’s population [16]. Although modern wheat varieties show relevant agronomic performances even in marginal environments, the landraces represent a valuable reservoir of important traits [17,18,19,20]. More recently, the evolution of durum wheat from ancient to modern varieties resulted in grain yield and gluten strength increases but a decrease in grain protein concentration. In turn, the quality of pasta improved, but with a decrease in its versatility.

Sicily, the southernmost region of Italy, is the second-largest durum-wheat-producing Italian region, with 343,500 hectares cultivated and 931,800 tons produced in 2021 [5]. Wheat arrived in Sicily around the 3rd millennium B.C. and found an ideal cultivation environment there, but it was only under Roman rule that Sicily became known as the ‘Granary of Italy’. Due to its position in the center of the Mediterranean, Sicily is characterized by a subtropical climate that is partially semi-arid, which played a crucial role as a crossroads in the evolutionary history of wheat by acting as a hinge (also in terms of food) between Asia, Africa, and Europe [21]. Over the centuries, a large wheat germplasm collection obtained from the selection of landraces adapted to areas of Sicily has been developed, some of which have been reproduced by farmers. The earliest historical document on Sicilian wheat varieties dates back to the 17th century, but their detailed descriptions proposed in the 20th century mainly are attributable to Ugo De Cillis [22]. ‘Timilia’ is described as one of the oldest Sicilian varieties [23]; it is currently cultivated throughout Sicily and is also known as ‘Tumminia’, ‘Triminia’, ‘Diminia’, and ‘Marzuolo’. It is still used to produce 30% of the local and traditional bread named ‘Castelvetrano black bread’, which is highly appreciated by consumers [24]. De Cillis described the ‘Russello’ variety, which is spread in the hilly hinterland throughout Sicily except for the western province of Trapani [22].

In addition to these two important local varieties, the Sicilian durum wheat germplasm includes other well-adapted landraces, the names of which have toponymic or morphological origins. Three similar varieties (‘Bidì’, ‘Margherito’, and ‘Cappelli’) originated from a North African population created by Nazzareno Strampelli in 1915.

Rivet wheat (*T. turgidum* subsp. *turgidum*) landrace group or so-called ‘Bufale’ are distinguished by color, all showing a mealy (softer) fracture of the kernel, a trait closer to bread than durum wheat [25]. Rivet wheat is widespread in the mountain range of the Calabria region, where durum wheat is not largely cultivated [26], and shows rusticity traits that are useful for growing in marginal soils and resistance to diseases [27]

‘Farro Lungo’ (*T. turgidum* ssp. *turanicum*, synonym ‘Perciasacchi’) was reported in Sicily at the beginning of the 19th century [28]. Later, De Cillis [22] described it as a typical cultivated wheat in Sicily.

Alongside many durum and tetraploid wheats, a few bread wheat landraces are still cultivated in Sicily; for example, ’Maiorca’ and ‘Cuccitta’ in the northeastern regions and ‘Maiorcone’ and ‘Romano’ mainly in the Catania plain.

Recently, due to a greater interest in traditional, healthy, and sustainable food products, an increase in hectarage dedicated to the cultivation of these ancient Sicilian wheat varieties has been observed. To regulate their seed production, a section in the National Variety Register for ‘Conservation Varieties’ has been established following Directive 98/95/EC [29]; to date, 27 durum wheat and 2 bread wheat Sicilian varieties were registered.

The need for identity conservation of local varieties stimulated research on their genotyping to both decipher the genetic diversity and develop a certification system. In the past, wheat genetic diversity was mostly evaluated using morphological and phenological descriptors [30,31,32]. More recently, different molecular markers have been utilized, such as single-nucleotide polymorphisms (SNPs) for marker–trait association [33,34], genetic diversity and population structure [35,36,37], genomic selection [38,39,40], and quantitative trait loci mapping [41,42]. Several studies focused on Italian wheat germplasm diversity analysis [43,44,45] using these markers [46,47,48].

Thanks to the activities of De Cillis and the preservation carried out by Sicilian farmers, a germplasm collection of Sicilian wheat is now conserved in the gene bank of the “Experimental Sicilian Station for wheat crop” (ESS) at Caltagirone (province of Catania, Italy). Over the last century, different populations of the same variety have been distinguished. Currently, the ESS contains more than 200 accessions of different *Triticum* species and subspecies, including durum and common wheat.

This study aimed to assess the morphogenetic diversity of the ESS germplasm collection using a high-density SNP array. In the analysis, durum and bread wheat landraces from the Calabria region were included that could also be conveniently preserved and exploited in future breeding programs. Interestingly, the genetic diversity of the EES collection was compared to the improved varieties of both old and modern wheat using the same SNP array while also defining their population structure. Moreover, the inter- and intrapopulation genetic variations in the historical Sicilian durum wheat varieties were also estimated. This research laid the foundation for expanding the genetic variability of cultivated wheat through the use of these underexplored southern Italian landraces.

## 2. Results

### 2.1. Genetic Relationships among Triticum Species in the Entire Dataset

The genetic diversity among *Triticum* species, which are widespread in Southern Italy (Sicily and Calabria regions), was investigated by using a high-throughput genotyping system based on a wheat 90k SNP array. After filtering, 20,899 high-quality SNPs out of 81,587 SNPs were retained (Appendix A).

A phylogenetic tree on the 126 wheat accessions from Sicily and Calabria was defined via a cluster analysis based on the UPGMA algorithm and the Nei’ genetic distance among the SNP-derived genotypes. Four main groups were identified: *T. aestivum*, *T. monococcum*, and two subspecies of *T. turgidum* (Figure 1A).

The first cluster included the *T. aestivum* accessions. Among these, the old varieties from different regions were separated from the modern varieties ‘Anapo’, ‘Anforeta’, and ‘Rossella’ (Appendix A). The second cluster was represented by the two *T. monococcum* accessions. The third cluster comprised all the rivet wheat accessions, the ‘Bufale’ group, and three Sicilian old varieties (‘Paola’, ‘Bivona Casedda’, and ‘Ciciredda’). All the durum wheat accessions and the more distant emmer wheat (*T. turgidum* ssp. *dicoccum*) landrace ‘Garfagnana’ and variety ‘Padre Pio’ comprised the fourth-largest cluster. In the latter, it was possible to distinguish several durum wheat subclusters (Figure 1A).

The more recently improved varieties (‘Claudio’, ‘Svevo’, ‘Core’, and ‘Duilio’) were grouped together; the older improved varieties (‘Trinakria’, ‘Senatore Cappelli’, and ‘Capeiti’) were clustered with the older varieties from North Africa (‘Tripolino’, ‘Bidì’, and ‘Margherito’). The turanicum wheat (*T. turgidum* subp. *turanicum*) gathered within this last subcluster maintained their distinctness except for the “Farro lungo 03” accession. The most mixed subcluster, which encompassed 84 Sicilian durum wheat accessions based on gene bank information, was originally classified into four groups (*durum*-T—‘Timilia’; *durum*-R—‘Russello’; *durum*-RI—‘Russello Ibleo’ or ‘Ruscia’, and *durum—*all the other durum wheat accessions analyzed here). The 14 *durum*-T samples and ‘Francesa_01′ accessions that clustered showed narrow genetic distances. Eleven *durum-*RI accessions were distinguished from the *durum-*R group except for ‘Sicilia Lutri’, Russello03, and Sicilia01. Four landraces from Sicily (‘Urria’, ‘Inglesa’, ‘Francesone’, and ‘Cannara’) were also included in this subcluster. Finally, the last subcluster included 11 *durum-*R group, Ruscia01, and 17 durum wheat landraces (*durum* group, Appendix A). Among these were ‘Gioia’ and ‘Vallelunga Glabra’ (synonym), ‘Giustalisa’ and ‘Sammaritana’, and ‘Niuru’ and ‘Castiglione Pubescente’ (similar to pairs).

Overall, moderate pairwise fixation index (Fst) values among the wheat species and subspecies were recorded (Figure 1B, Appendix A). The highest *Fst* value was observed between the diploid *T. monococcum* and the two-tetraploid subsp. *dicoccum* and *turanicum* (0.292 and 0.300, respectively). Instead, a low differentiation (<0.17) was detectable in the other tetraploid subspecies (rivet wheat and ‘Timilia’ and ‘Russello Ibleo’ durum groups), with the lowest value (0.025) highlighted between the pair *T. turgidum* subsp. *durum* and the Russello group (R). Among the four durum wheat groups previously described, a low *Fst* (≤0.07) was evident.

The PCoA analysis (Appendix A) clustered the wheat collection into six distinct groups according to the first (PCoA1 ≈  31% of genetic variance) and the second (PCoA2 ≈ 10% of genetic variance) principal coordinates. The durum wheat group T and the rivet wheat accessions formed two separate clusters. In contrast, the other durum accessions, including the groups R and RI, as well as the *turanicum* wheat accessions, were gathered in a unique cluster. The population structure of the 126 *Triticum* accessions was inferred using fastSTRUCTURE, which did not account for linkage disequilibrium (LD) between genetic markers (Figure 2). The best number for model complexity (*K*) was between 4 and 6 when using the algorithm for multiple choices *chooseK.py*, with *K* = 6 as the model used to explain the structure in the dataset (Appendix A). Instead, the prediction error approach recorded the optimum pools at *K* = 5 (Appendix A). Therefore, the lowest model complexity within the range was explained at this value (*K* = 5). The inferred population membership coefficients are shown in Appendix A. A total of 110 accessions (87%) were grouped into 5 gene pools; the remaining 16 genotypes were shown to have an admixture genetic structure using a membership probability threshold of 0.7 (Figure 2, Appendix A). The structure analysis confirmed that Francesa01 belonged to the T group (*K*5), in agreement with the phylogenetic (Figure 1A) and PCoA analyses (Appendix A). Among others, five Sicilian accessions (‘Tripolino’, ‘Tunisina’, ‘Chiattulidda’, ‘Pavone,’ and ‘Vera’) showed an admixture structure. In addition, Ruscia01 and ‘Duro SG3′, the varieties obtained by the cross Timilia x ‘Biancuccia’ (Appendix A), as well as the *dicoccum* ‘Garfagnana’, also showed an admixture structure. All of the bread wheat accessions were included in the same membership group (*K4*), while the rivet wheat accessions were in *K1* (Appendix A). Improved varieties of durum wheat and the two landraces ‘Bidì ‘and ‘Margherito’ were grouped in *K2* together with Farro Lungo 03, a Sicilian *turanicum* wheat, and ‘Padre Pio’, a variety obtained from a cross durum wheat x emmer. All of the remaining Sicilian durum wheat accessions, including the R and RI groups, were assigned to cluster *K3* (Figure 2, Appendix A).

### 2.2. Morphological Diversity of Triticum Landraces

Morphological characterization was performed on a subset (110) of the wheat germplasm under study. Sixteen accessions were not included in the analysis based on the twenty-four International Union for the Protection of New Varieties of Plants (UPOV) descriptors (Appendix A) because only the germplasm representative of ancient local varieties was considered.

The clustering pattern of the morphological traits was calculated based on the Euclidean distance (Figure 3). All of the bread wheat accessions were grouped into one cluster except for the two Calabrian landraces (‘Rosia’ and ‘Incensarola’), which showed a unique profile that was different from most of the Sicilian accessions, mainly due to plant-height-related traits (var7 and var10) and the absence of glaucosity on the lower leaf blade side (var4). In addition, the presence of awns (var9) was shown by both ‘Maiorca’ from Calabria and two Sicilian ‘Maiorcone’ accessions but was absent in the other Sicilian common wheat accessions. The two Sicilian *T. turgidum* subsp. *turanicum* accessions clustered together with ‘Turanicum_PI283795′, an accession from Afghanistan included in the ESS collection since 2012, and ‘Giustalisa’, a Sicilian landrace of durum wheat.

Another cluster included all seven Calabrian rivet wheat accessions together with BufalaSB02, while the other Sicilian rivet wheat accessions were distributed in different clusters. Six accessions included in the *durum-*R group were clustered together with two durum wheat accessions (VallelungaGL02 and CastiglionePUB01). The remaining accessions of the *durum-*R group were distributed in other clusters together with the durum wheat accessions belonging to the RI and T groups. A slight coloration of the ear characterized the RI and R groups. All of the T group accessions showed a pyramidal ear shape.

A principal component analysis (PCA) of the standardized data was applied to display the wheat morphological trait relationships and their application in germplasm characterization (Appendix A). The first two components explained only 29.1% of the total variation among the morphological traits investigated. However, the first dimension was able to clearly separate the samples belonging to *T. turgidum* subsp. *turanicum* (top-left quadrant) and *T. aestivum* (top-right quadrant), with var3, var5, and var 6 (Appendix A) showing the strongest influence (cos^2^ = 0.60) (Appendix A). Likewise, almost the entire germplasm of *T. turgidum* subsp. *turgidum* studied (11 out of 14 samples; 79%) was included in the bottom-left quadrant. Of the traits, var7 and var10 were the most influential components (Appendix A). Overall, the flag leaf glaucosity of the sheath (var3), ear glaucosity (var6), and plant length (stem, ear, awns, and scurs; var10) were shown to strongly influence PC1 and PC2.

A Pearson correlation matrix showed that variables 3, 4, and 5 (glaucosity traits) were highly correlated in a positive manner, as were var 9 and var10 (ear and plant length) (Appendix A). These were grouped into one of the three main clusters of traits (Figure 3).

### 2.3. Population Genetics and Diversity of Sicilian Historical Varieties Belonging to Durum Wheat

The genetic background of the samples that belonged only to *T. turgidum* subsp. *durum* (84 accessions) were then evaluated first through an evaluation of their main genetic parameters.

The *H_e_*, *H_o_*, AR, and *Fis* values were calculated (Table 1), which classified the samples into four groups based on the membership assigned after the genetic (Appendix A) and morphological evaluation (Appendix A). A low *H_e_* was recorded in all four groups that ranged from 0.13 (T) to 0.17 (old durum varieties), as well as a low *H_o_* that ranged from 0.15 (old durum varieties) to 0.20 (RI). Allelic richness (AR) was largely uniform in all four groups, with the highest value recorded by the old durum varieties (Table 1). The *Fis* values, which described the genetic diversity within each group, were negative for all groups except for *T. turgidum* subsp. *durum* (0.130) and ranged from −0.037 (*T. turgidum* subsp. *durum*-R) to −0.301 (*T. turgidum* subsp. *durum*-RI), which highlighted a slight heterozygous excess.

The genetic relationships among the Sicilian durum wheat accessions and the population structure were analyzed using the NJ and DAPC analyses, respectively. Minimum spanning networks (MSNs) were also computed for the same subpanel. The Bayesian information criterion (BIC) method allowed six clusters as the optimal number of groups (Appendix A). Membership values of each accession to the six clusters are available in the assigned plot (Appendix A). The DAPC scatter plot showed a triangle-shaped distribution, which highlighted that pool 6 (yellow) was divergent from the others (Figure 4). This pool mainly encompassed accessions belonging to the T. turgidum subsp. durum (26%, 9 out of 35 genotypes), including the modern durum wheat varieties (‘Capeiti’, ‘Claudio’, ‘Core’, ‘Duilio’, ‘Simeto’, ‘Svevo’, and ‘Trinakria’), some very old landraces/varieties (‘Tripolino 01′ and ‘Tripolino 02′), and the ‘Vera’ accession of the durum-R group (Appendix A). The other accessions of the T. turgidum subsp. durum group were mainly split between three pools. These were pool 1 (dark green), which grouped the old varieties ‘Bidì’, ‘Margherito’, and ‘Senatore Cappelli’; pool 4 (deep pink), which mainly encompassed the accessions of the durum-R group (75%, 15 out of 20 genotypes); and pool 3 (blue), which represented 69% of this pool (11 out of 16 samples). The other two groups—durum-RI and durum-T—were split between two main pools: pool 5 (light green) and pool 2 (orange) (Appendix A). In pool 5, three durum wheat varieties (‘Francesone’, ‘Inglesa’, and ‘Urria’) and three accessions previously classified in the durum-R group (Sicilia01, ‘Sicilia Lutri’, and ‘Cannara’) were also included. Interestingly, the ‘DuroSG3′ sample, which was selected from a cross between ‘Timilia’, the old variety ‘Biancuccia’ belonging to the durum-T group, and the durum-RI accession ‘Ruscia 01′, were included in Group 3 (Figure 4 and Appendix A).

Finally, the two accessions ‘Castiglione GL03′ and ‘Castiglione PUB01′ together with the old varieties ‘Gioia’ and ‘Niuru’ were included in pool 4, while the accession ‘Francesa01′ was included in pool 2, which was in agreement with the UPGMA cluster analysis. In contrast, the accession ‘Francesa02′ was assigned to pool 3 together with the old durum wheat varieties (Figure 4).

The neighbor-joining (NJ) tree, which included only the accessions belonging to T. turgidum subsp. durum, highlighted a large cluster that gathered all improved durum wheat varieties and the landrace ‘Vera’ (Figure 5A). The remaining accessions belonging to the durum group were split into several clusters that mainly also included genotypes of the durum-R group. Particularly, the durum-R group was largely shifted into two clusters: one shared with ‘Giustalisa’, ‘Lina’, ‘Gigante’, and ‘Sammaritana’, and four old durum wheat varieties, and the other with ‘Girgentana’ and ‘Martinella’) (Figure 5A). In contrast, all accessions of the durum-RI group except Ruscia01 were clustered in a private branch that also included three old Sicilian varieties (‘Francesone01′, ‘Inglesa02′, and ‘Urria01′) and three accessions of the durum-RI group (‘Sicilia02′, ‘Sicilia Lutri’, and ‘Cannara’), which were previously assigned to the durum group, in agreement with the population structure analysis (Figure 4 and Appendix A). Finally, the durum-T group formed a distinct cluster that included all of the ‘Timilia’ accessions together with the ‘Francesa01′ (in agreement with the DAPC) and ‘Tunisina01′ accessions (Figure 5A).

The minimum spanning network (MSN) confirmed the NJ tree results (Figure 5B), with accessions of the *durum* group (dark green) widespread in the network and admixed mainly with *durum*-R samples, which showed a close relationship (Figure 5B). In contrast, the accessions belonging to *durum*-RI (blue) and *durum*-T (deep pink) appeared in distinct nodes and had a greater diversity than the other groups (Figure 5B).

The multidimensional scaling (MDS) plots (Appendix A) illustrated distinct groups within the durum wheat class. This suggested that a portion of the genetic diversity from the accessions of *durum*-R and *durum*-RI groups had been introgressed in the remaining durum landraces/varieties. In contrast, the *durum*-T group accessions’ gene pool was shown to remain underutilized by breeding.

## 3. Discussion

The presence of wheat in the Mediterranean basin dates to 7000 BP (before the present) [49]; later, wheat spread to North Africa (Tunisia, Morocco, and Algeria) and southern Italy, including Sicily, where it was spread throughout the island. Here, starting from the 3rd millennium BC, wheat found the best environmental conditions for adapting and evolving. The process of both natural and human selection, as well as the settlement of different populations in Sicily, led to the introduction and/or selection of a large number of local wheat varieties. Therefore, the genetic diversity of *Triticum* landraces cumulated across the centuries and reached a maximum at the beginning of the previous century when breeders began the distribution of new improved varieties [50]. This wide range of genetic variability available in Sicily was described in detail by Ugo De Cillis [22], although the descriptions of some ancient varieties can be found in 17th-century publications [51].

From 1929 until 1948, De Cillis headed the “Experimental Sicilian Station for wheat crop” (ESS) at Caltagirone (province of Catania in Sicily), which collected information on the morphological, physiological, and technological traits of Sicilian wheat landraces/varieties. The first description/classification based on the Percival method [52] placed some limitations on bringing the observed genetic stock into the collection, forcing De Cillis to purify some accessions using recurrent selection and to assign names according to their area of origin. Nowadays, 39 5old varieties of the original core collection of De Cillis’ work have been maintained and incremented by the ESS. This collection represents the large biodiversity available in Sicily since 1900 and is to be preserved as an untapped reservoir for expanding the genetic structure of modern varieties. Furthermore, the growing awareness of the importance of a healthy lifestyle is increasingly linked to the consumption of sustainable alternative foods such as old varieties, which can reduce the environmental burden of human food production. This heightened interest in “ancient grains” was needed to regulate their production and multiplication at the national and regional levels. In turn, this necessitated the development of protocols to avoid commercial fraud and to sustain the economic profits derived from the wheat food chain [29].

Many studies were carried out to characterize and identify wheat species accessions based on morphological characterization according to UPOV traits [53,54,55], which are, unfortunately, influenced by the environment. In those studies, a relatively large variation based on morphological traits was detected among accessions.

The development of the first molecular marker classes furnished novel tools for avoiding misclassification due to the environment. Among others, the simple sequence repeats (SSR) represent more informative molecular markers that are largely detectable in plant genomes. The genetic diversity of several wheat germplasm collections has been evaluated and distinguished species, subspecies, and genotypes [48,56,57,58]. Lastly, with the advent of next-generation sequencing (NGS) techniques, a very large panel of SNPs has been set up as a useful tool to provide a detailed reproducible genetic fingerprint of the wheat germplasm [59,60,61]. SNPs are point mutations that result in single-base-pair divergence among DNA sequences present in both genome coding and noncoding regions. These are also utilized for genetic characterization in wheat, in which the tetraploid and hexaploid conditions made the utilization of microsatellites more difficult.

Several studies assessed the genetic diversity of the Sicilian wheat germplasm described by De Cillis using both SSR [25] and SNP [47,48,62]. Here, we assembled a definitive panel of 126 wheat accessions to assess their genetic relatedness using a large SNP panel. In the collection, bread and durum wheat landraces and varieties (old and modern), *T. monococcum* (EES collection), and many subspecies related to *T. turgidum* (*dicoccum, durum, turanicum,* and *turgidum*) were included. Our results confirmed the ability of SNP technology to easily distinguish *Triticum* species and subspecies. In addition, we were able to classify 84 durum wheat accessions (including old and modern varieties, as well as landraces) for genetic distances and structures in order to assess their genetic relationships. Moreover, three groups/populations related to ‘Timilia’, ‘Russello’, and ‘Russello Ibleo’ were also included, thereby also defining the intrapopulation genetic variability, as recently reported by Taranto et al. [63].

A study of 56,342 hexaploid accessions from CIMMYT and ICARDA germplasm banks showed that a large portion of genetic variability of hexaploid wheat varieties has not been sampled in breeding selection for modern varieties [64]. In our study, the genetic distinctness between the ancient bread wheat varieties and two improved varieties (‘Anapo’ and ‘Anforeta’) confirmed and suggested a useful gene reserve in the EES collection to be explored and subsequently used in future breeding programs. In the bread wheat cluster, a selected line of the *T. aestivum* subsp. *spelta* and four ancient bread wheat landraces collected in Calabria were also included.

The distinctness between the 14 rivet wheat accessions and the other three tetraploid subspecies, as previously reported [25,62], was confirmed, which was in agreement with De Cillis’ description of the landraces ‘Bufala Rossa Corta’, ‘Bufala Rossa Lunga’, ‘Bufala Nera Corta’, ‘Bufala Nera Lunga’, ‘Ciciredda’, ‘Bivona’, and ‘Paola’ [22]. These underutilized rivet wheat genotypes represent a significant genetic stock that could be useful for maintaining wheat cultivation in specific rural mountainous areas of Sicily and other Mediterranean regions such as Calabria, which are usually considered marginal and unsuitable for élite durum wheat cultivation. In addition, a private cluster related to the rivet wheat accessions (a subcluster from Calabria and another from Sicily) was revealed for the first time, in contrast to Mangini et al. [59]. The UPGMA cluster analysis comprised all the durum wheat accessions in the largest cluster, including four accessions belonging to *T. turgidum* subsp. *turanicum* and, more distant, the two emmer wheat varieties: ‘Garfagnana’ (a landrace cultivated in Tuscany, Italy) and the variety ‘Padre Pio’. This result did not agree with Ganugi et al. [62], who reported three separated clusters for *turanicum*, *dicoccum,* and *durum* accessions.

Khorasan wheat (*T. turgidum* subsp. *turanicum*), better known under the brand Kamut, has recently drawn a large commercial interest due to its high protein content and valuable nutritional qualities [65]. However, there is limited information available on the partitioning of the genetic diversity on the available accessions of Khorasan/*turanicum* wheat. It was reported that they are related to durum wheat [66]. Hence, significant variations in the agronomic traits have been recorded among Khorasan accessions [67]; a univariate analysis was not able to clearly distinguish *turanicum* and *durum* wheat varieties [68], as was also demonstrated in our study. Only the accession Farro lungo 03 showed a posterior probability value close to 1 (0.99998) in the same membership group of durum wheat accessions, while the remaining three accessions, including the ‘Khorasan’ commercial variety and TuranicumPI283795, showed an admixture genetic structure (K2 and K3).

The genetic analysis was then intensified and limited to the *T. turgidum* subsp. *durum* accessions, which underlined a higher observed heterozygosity in the three R, RI, and T groups compared to the group that included the more recent improved varieties, as could be expected. This observation might infer a mixing of previously isolated populations. The inbreeding coefficient (*Fis*) value was close to 1 for the more recent durum wheat varieties (0.130), while the negative values recorded for the T, RI, and R groups highlighted a slight heterozygous excess. Both the ‘Timilia’ and ‘Russello’ accessions belonged to the two ancient wheat varieties, which were widely cultivated in Sicily during the 18th and 19th centuries, respectively. Thereafter, the cultivation of these two accessions spread to many Sicilian areas. The accessions belonging to the RI (‘Russello Ibleo’) group remained confined to their area of origin (Monti Iblei) and appeared to be distinguished by ‘Russello’ accessions. Recently, intra- and interpopulation genetic variations of two large collections of ‘Russello’ and ‘Timilia’ were explored by Taranto et al. [63]. In agreement with our results, a low genetic distance among the seven populations of ‘Timilia’ was observed, while higher values among the three ‘Russello’ populations were measured. The same authors identified two different groups within the ‘Russello’ populations in accordance with their geographical origin: ‘Monti Iblei’ in southeastern Sicily (named ‘Ruscia’) and central-western Sicily. Furthermore, the MSN confirmed the UPGMA and NJ tree results for the genetic relationships among the Sicilian durum wheat ancient varieties and *durum*-R samples, while *durum*-RI (blue) and *durum*-T (deep pink) appeared in distinct nodes with greater diversity than other groups. These results agreed with those previously reported by Taranto et al. [46].

Southern European crops after the 2000s have been subjected to high temperatures in summer, early drought, and very hot temperatures at heading, exacerbating the risk of yield loss [69]. Van de Wouw et al. [70,71] reported that genetic erosion in terms of loss of alleles was not observed in wheat before the 1990s. A strengthening of crop resilience can be found within the genetic variability present in landraces and old wheat varieties [44,72,73]. The morphogenetic analysis of the Sicilian *Triticum* germplasm collection maintained at ESS highlighted the importance of preserving this precious reservoir of genetic resources that have developed and evolved over the centuries in the Mediterranean environment.

## 4. Materials and Methods

### 4.1. Plant Materials

The germplasm collection included 126 accessions: 2 hexaploid wheats, *T. aestivum* L. (19 accessions), and *T. aestivum* subsp. *spelta* L. Thell. (one); the cultivated einkorn *T. monococcum* L. subsp. *monococcum* (two); four tetraploid subspecies of *T. turgidum* (104) consisting of 14 rivet wheat accessions (seven from Sicily and seven from Calabria); four *T. turgidum* subsp. *turanicum* accessions; two *T. turgidum* subsp. *dicoccum* accessions; 75 Sicilian landraces and nine improved varieties of durum wheat (Appendix A). The durum wheat germplasm under study comprised 39 accessions of the original nucleus collected and characterized by De Cillis in 1927, which was partially maintained at the ESS at Caltagirone (province of Catania, Italy). Other accessions were retrieved from the main international germplasm banks (USDA, CGN, IPK, and IHAR). Instead, several accessions of the three historical Sicilian varieties Sicily (‘Russello’, ‘Timilia’, and ‘Russello Ibleo’/’Ruscia’), which are among the most widespread on the island, have been preserved in the ESS germplasm bank over the last 20 years. All ESS accessions were multiplied in the experimental field of Caltagirone, province of Catania, Sicily (37°05′58′′ N, 14°29′56′′ E, 280 m a.s.l.). The germplasm collection from Calabria (seven rivet wheat and four common wheat), which was derived from old local populations, is maintained in the collection at the Department AGRARIA of the University Mediterranea of Reggio Calabria—Italy, in the experimental field of the “Casello” Agricultural Experiment Centre of the Regional Agency for Agriculture (ARSAC), which is located in San Marco Argentano, Calabria (38°10′24′′ N, 15°45′12′′ E, 232 m a.s.l.) in a classic cereal-growing area north of Cosenza.

### 4.2. Bio-Morphological Traits

The bio-morphological characterization included 110 of the 126 genotypes under study. Sixteen genotypes that were not representative of the local varieties from Sicily and Calabria were excluded from this analysis (Appendix A).

Twenty-four UPOV bio-morphological traits [74] were measured for each accession (Appendix A). The measurements were carried out in the field during the vegetative stages using the Zadoks scale system [75]: from stage 10 (first leaf passing through the coleoptile) to stage 60 (full growth but not flowering) and in the laboratory using a representative sample of 20 ears.

### 4.3. Statistical Analysis for Morphological and Quality-Related Traits

Samples of wheat grains were analyzed for 24 main morphological traits (Appendix A). To easily visualize the differences between samples, a heatmap was developed through R/pheatmap (https://CRAN.R-project.org/package=pheatmap; accessed on 13 July 2022). A principal component analysis (PCA) was also performed using the R package FactoMiner [76]. Finally, the Pearson correlation coefficient (*p* < 0.05) was also calculated among the morpho- and quality-related traits using the R/Hmisc package (https://cran.r-project.org/web/packages/Hmisc/index.html; accessed on 13 July 2022). A scatter plot with the correlation coefficients and their significance was developed using the PerformanceAnalytics package in R (https://cran.r-project.org/web/packages/PerformanceAnalytics/index.html; accessed on 13 July 2022).

### 4.4. DNA Extraction and SNP Genotyping

Genomic DNA was extracted from fresh leaves of young seedlings of each variety. The GenElute Plant Genomic DNA Miniprep Kit (Sigma-Aldrich, St. Louis, MO, USA) was used. DNA quantity and quality were checked via both electrophoresis (1% agarose gel) and a NanoDrop^®^ND-1000 (Thermo Scientific, Waltham, MA, USA).

Two hundred nanograms of genomic DNA were used as template for each reaction in the wheat 90k SNP array [77] for genotyping, following the manufacturer’s instructions (Illumina Inc., San Diego, CA, USA). Raw data were visualized and analyzed using the GenomeStudio V2011.1 software (Illumina). The dataset was filtered and standardized as reported in Mercati et al. 2021 [78]. SNPs with an NA rate >1%, a MAF <5%, an NA rate for each individual >20%, and a polymorphic rate between individuals <20% were excluded by subsequent analyses. After filtering, 20,899 high-quality SNPs out of 81,587 SNPs provided were retained (Appendix A).

### 4.5. Cluster, PCoA, and Fst Analysis

Phylogenetic and PCoA analyses were performed to estimate the overall relationship among varieties while considering the entire wheat germplasm investigated (126 genotypes) and the samples only belonging to *T. turgidum* subsp *durum* (84) collected. The distance-based dendrogram with Nei’s genetic distance [79], unweighted pair group method with arithmetic mean (UPGMA), neighbor-joining (NJ) algorithm (bootstraps based on 1000 resamplings), and PCoA were developed using R/poppr [80] and R/adegenet [81], respectively, and visualized using R/ggtree [82]. The Wright’s population pairwise fixation index (*Fst*) [83] was computed among species and groups using R/HierFstat [84]; a heatmap based on *Fst* values was developed using R/gplots (https://CRAN.R-project.org/package=gplot; accessed on 13 July 2022).

### 4.6. Population Structure Analysis

The number of genetic pools (*K*) was computed for the entire wheat dataset using an admixture model performed through fastSTRUCTURE [85] using the .bed, .bim, and .fam as input files generated by PLINK v1.90 [86] and the structure.py Python script. The algorithm for multiple choices (chooseK.py) was applied to choose the best number for the model complexity (*K*) by evaluating the model complexity that maximized the marginal likelihood and model components used to explain structures in the data. The prediction error for each *K* was also computed using the cross-validation (*--cv*) function to extract the optimum *K*. A Bayesian analysis was also carried out for the *T. turgidum* subsp *durum* subset (84 samples) using a discriminant analysis of principal components (DAPC) implemented in R/adegenet. The Bayesian information criterion (BIC) method [87] was used to infer the *K-*means clustering. The analyses were performed independent of the membership group.

### 4.7. Population Genetics and Diversity of Sicilian Varieties Belonging to T. durum

The filtering procedure and LD (linkage disequilibrium) pruning (r2 = 0.20) resulted in datasets of high-quality SNP markers that were used to evaluate in detail the genetic diversity of samples belonging to *T. turgidum* subsp *durum*. The main genetic parameters (*Ho*, *He*, *Fis*, and allelic richness (AR)) were evaluated using the R/OutFLANK package [88]. A pairwise identical-by-state (IBS) distance among all individuals using R/SNPRelate [89] was calculated and a multidimensional scaling (MDS) analysis was performed. In addition, to evaluate the genetic distances among the varieties gathered in the four groups (Appendix A) of *T. turgidum* subsp *durum*, *T. turgidum* subsp *durum*-R (Russello), *T. turgidum* subsp *durum*-RI (Russello Ibleo), and *T. turgidum* subsp *durum*-T (Timilia), a minimum spanning network (MSN) was also developed in R/poppr [80].

## 5. Conclusions

In conclusion, SNPs represented powerful tools for exploring genetic diversity in *Triticum* germplasm. The relationships among old Southern Italian landraces, populations, and old and modern varieties were confirmed. We developed a definitive panel of 126 accessions that were mainly preserved at the ‘Experimental Sicilian Station (EES) for Granicoltura’, including the collection derived from the Nazareno Strampelli research activity and described by De Cillis, among which many landraces and old varieties were cultivated in Sicily and Southern Italy since the beginning of the 20th century. They represent an important reservoir of genes/alleles that can be used in durum and bread wheat breeding to counteract the effects of climate change. The adopted germplasm collection and SNP array allowed us to distinguish not only the *Triticum* species (as was expected), but also many *T. turgidum* subspecies. Interestingly, we were able to distinguish the subspecies *dicoccum* and *turgidum* from most cultivated *durum*. However, the subspecies *turanicum*, including the commercial cultivar Khorasan, was not distinguishable. In addition, two Sicilian old landraces/populations, namely ‘Timilia’ and ‘Russello’ and its variant ‘Russello Ibleo’, were characterized among the 84 durum wheat accessions (landraces and old and modern varieties). The old Sicilian landraces/populations were clearly clustered and distinguished from the remaining durum wheat accessions, including the ‘Russello Ibleo’ variant. The present research confirmed the importance of the broad genetic base available not only in the Sicilian landraces, but also among the old varieties cultivated in Southern Italy over the last century and recently re-utilized for their high-quality traits. The interpopulation genetic diversity observed for ‘Timilia’ and ‘Russello’, together with the synonyms and misclassifications of some old wheat varieties, should be taken into account when registering old varieties in the National Registers of Conservation Varieties. By so doing, it will be possible to avoid commercial fraud and contribute to sustaining the profits of farmers that cultivate landraces and ancient varieties.

## Figures and Tables

**Figure 1 ijms-23-13378-f001:**
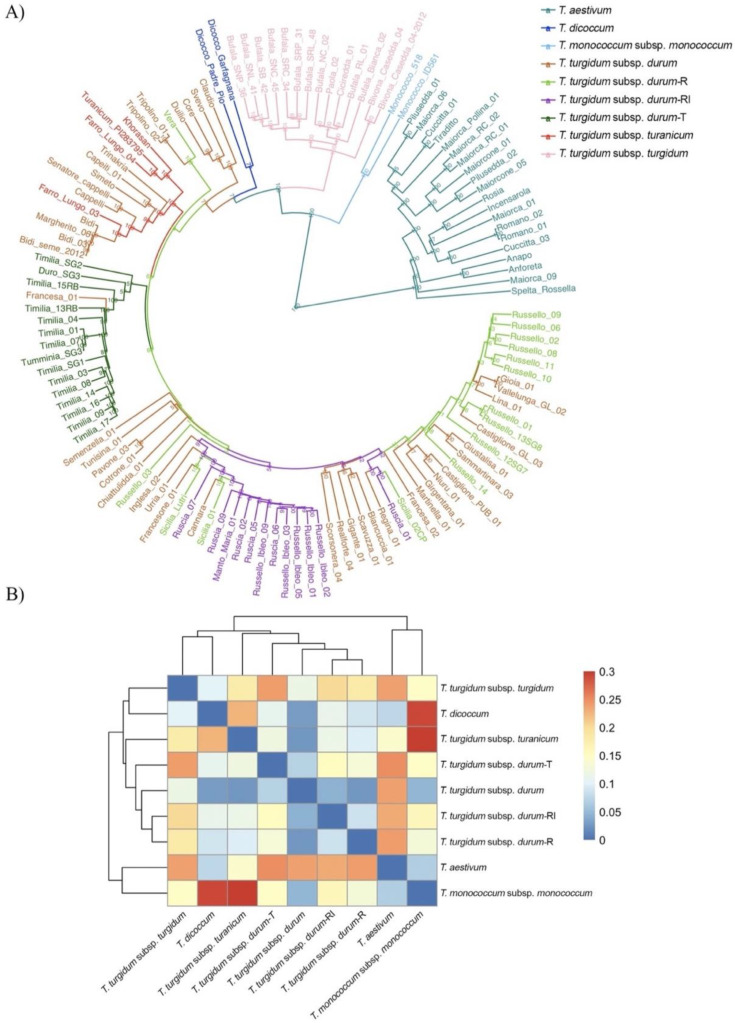
Genetic diversity of 126 wheat genotypes analyzed in the 90k SNP array. (**A**) UPGMA dendrogram. The varieties are highlighted based on their species and membership group (Appendix A). (**B**) Heatmap based on the population pairwise fixation index (*Fst*) evaluated among the wheat species and groups. R: Russello; RI: Russello Ibleo; T: Timilia.

**Figure 2 ijms-23-13378-f002:**
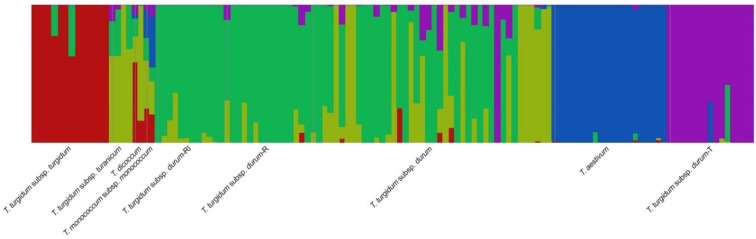
Population structure in the entire wheat dataset (126) as analyzed in the 90k SNP array. The optimum pool number was recorded at *K* = 5 by fastStructure. Each sample is represented by a vertical bar. The color proportion for each bar represents the posterior probability of assignment of each variety to one of the five groups of genetic similarity and ranges from 0 to 100% of the assignment probability range (Appendix A).

**Figure 3 ijms-23-13378-f003:**
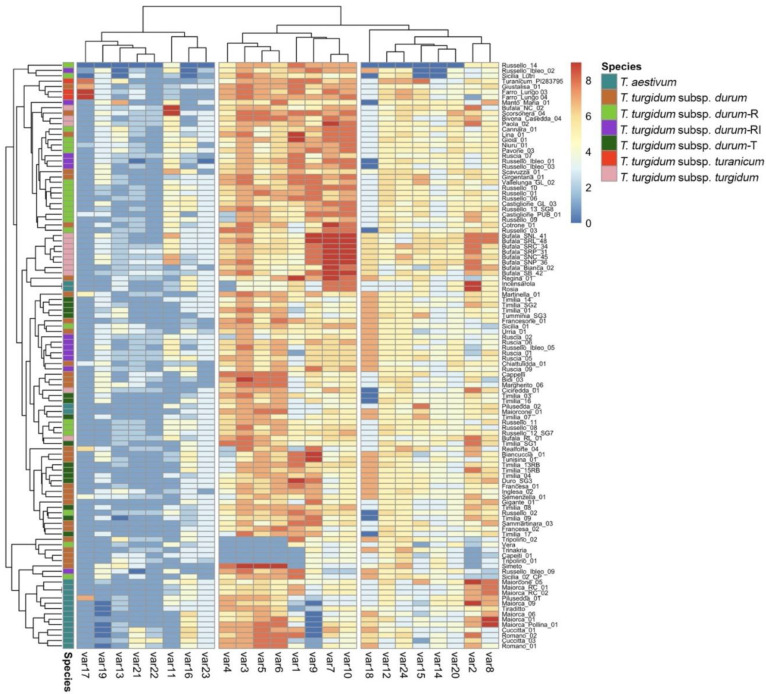
Heatmap of the morphological traits evaluated on a wheat germplasm subset (110) (Appendix A). R: ‘Russello’; RI: ‘Russello Ibleo’; T: ‘Timilia’. The colors in the heatmap indicate the values reported in Appendix A for each trait evaluated.

**Figure 4 ijms-23-13378-f004:**
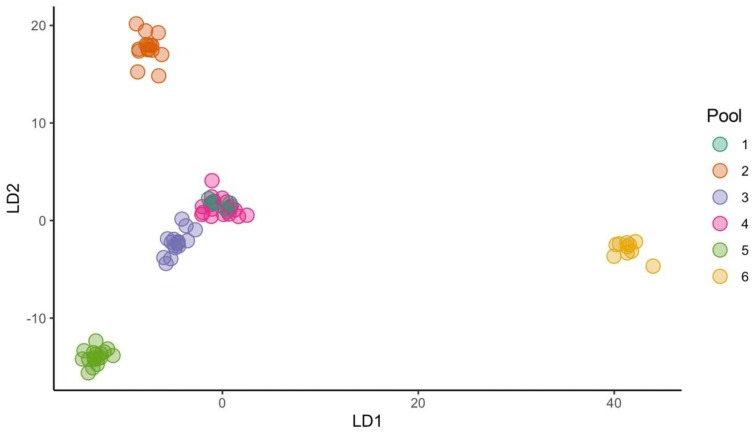
DAPC scatter plot of the 84 accessions belonging to T. turgidum subsp. durum (Appendix A) based on the first and second linear discriminants (LDs) at the best K (6) as evaluated using the BIC method (Appendix A). The membership for each sample is highlighted in Appendix A.

**Figure 5 ijms-23-13378-f005:**
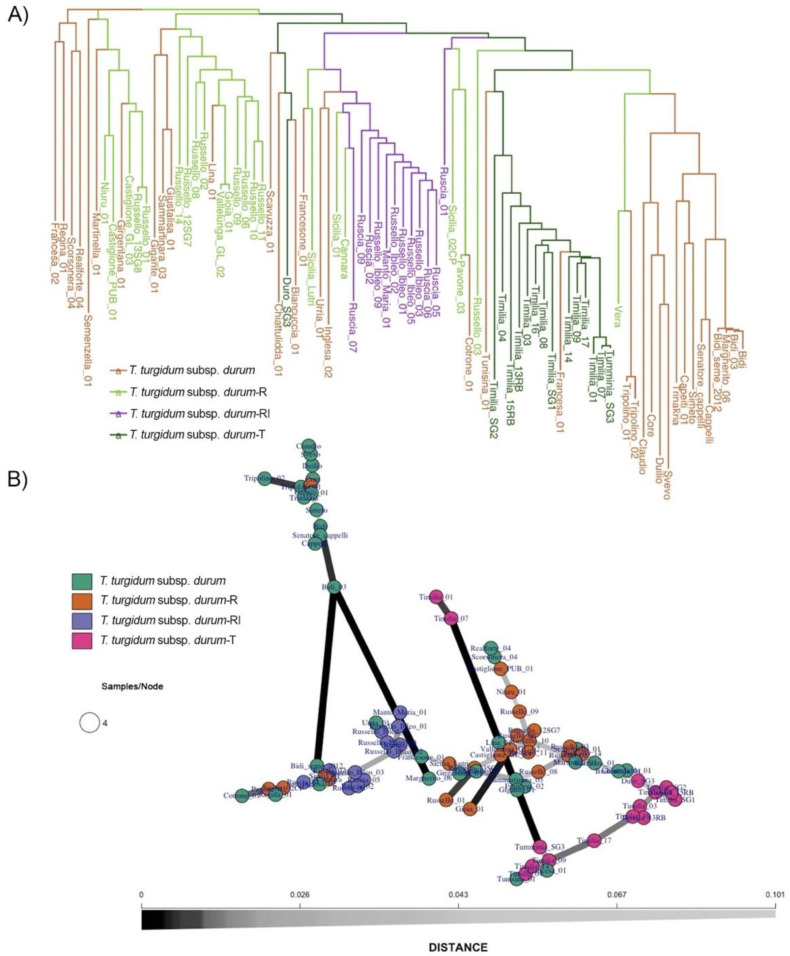
(**A**) NJ tree based on SNP profiles of varieties belonging to *T. turgidum* subsp. *durum*: *T. turgidum* subsp. *durum*, *T. turgidum* subsp. *durum*-R (‘Russello’), *T. turgidum* subsp. *durum*-RI (‘Russello Ibleo’), and *T. turgidum* subsp. *durum*-T (‘Timilia’). (**B**) A minimum spanning network (MSN) showing the multilocus genotypes (MLG) of accessions belonging to *T. turgidum* subsp. *durum* subset. The distances between the MLGs indicates Nei’s genetic distance; each MLG is a node and the genetic distance is represented by the edges. The nodes are connected by the minimum distance between samples, which allowed for reticulations. The grey scale of the edges is proportional to the genetic distance; the lines become darker for more-related nodes.

**Table 1 ijms-23-13378-t001:** Genetic parameters evaluated for the *T. turgidum* subsp. *durum* subset split into four groups: *T. turgidum* subsp. *durum* (all the other accessions), *T. turgidum* subsp. *durum*-R (‘Russello’), *T. turgidum* subsp. *durum*-RI (‘Russello Ibleo’), and *T. turgidum* subsp. *durum*-T (‘Timilia’).

Group	*H_o_*	*H_e_*	*AR*	*Fis*
*T. turgidum* subsp. *durum*	0.15	0.17	1.167	0.130
*T. turgidum* subsp. *durum*-R	0.16	0.14	1.142	−0.037
*T. turgidum* subsp. *durum*-RI	0.20	0.14	1.138	−0.301
*T. turgidum* subsp. *durum*-T	0.18	0.13	1.131	−0.199

*H_o_* = Observed heterozygosity; *H_e_* = expected heterozygosity; *AR* = allelic richness; *Fis* = inbreeding coefficient.

## Data Availability

The data presented in this study are available in the text or Appendix A here.

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
