# Peer review of "Elucidating the Genetic Relationships on the Original Old Sicilian Triticum Spp. Collection by SNP Genotyping"

_ijms, 2022, doi:10.3390/ijms232113378_

Round 1

Reviewer 1 Report

The manuscript is highly arranged and well illustrated story, but some comments should be taken into consideration before bring accepted. 

Arrange the keywords in alphabetic order. 

The introduction should be summarized. 

The aim should be improved. 

The resolution of Figure 1a should be improved. 

Author Response

The manuscript is highly arranged and well illustrated story, but some comments should be taken into consideration before bring accepted. 

Answer: Thank you for your positive evaluation. The manuscript has been modified following your suggestions. All changes are highlighted in red

- Arrange the keywords in alphabetic order. 

Answer: although IJMS does not specifically require keywords in alphabetical order, we arranged them in alphabetic order as suggested.

- The introduction should be summarized. 

Answer: thank you for your suggestion. The introduction has been summarized.

- The aim should be improved. 

Answer: the aim has been improved as suggested.

- The resolution of Figure 1a should be improved. 

Answer: the figure has been improved and included in the new version of the MS re-submitted

Reviewer 2 Report

This study evaluated the genetic diversity of approximately 126 Sicilian and Calabrian wheat lines. Using SNP arrays, the authors performed various genetic and genomic analyses and fairly well distinguished the triticum species. Overall, this manuscript is written well and clear. However, I have the comments as follows.

The raw SNP set consists of 81,587 SNPs, whereas the filtered dataset only contains 20,899 SNPs. A huge amount of data were discarded. This is rarely seen in diversity studies. The discarded data potentially contains lots of unique genetic information such as rare alleles. The authors indeed mentioned the data filtering criteria. It is surprising than such high amount of raw data were discarded. Did the authors try relatively loose data filtering criteria and run the analysis? In addition, most open access journals now require data availability. I suggest the authors deposit their raw data on NCBI or other public repositories.

The abstract needs to be improved and should clearly indicate the results like how many sub-populations were detected in the whole panel and the small panels used and what inference was deduced.

Author Response

This study evaluated the genetic diversity of approximately 126 Sicilian and Calabrian wheat lines. Using SNP arrays, the authors performed various genetic and genomic analyses and fairly well distinguished the triticum species. Overall, this manuscript is written well and clear. However, I have the comments as follows.ù

Answer: Thank you for your positive evaluation. The manuscript has been modified following your suggestions. All changes are highlighted in red

- The raw SNP set consists of 81,587 SNPs, whereas the filtered dataset only contains 20,899 SNPs. A huge amount of data were discarded. This is rarely seen in diversity studies. The discarded data potentially contains lots of unique genetic information such as rare alleles. The authors indeed mentioned the data filtering criteria. It is surprising than such high amount of raw data were discarded. Did the authors try relatively loose data filtering criteria and run the analysis? In addition, most open access journals now require data availability. I suggest the authors deposit their raw data on NCBI or other public repositories.

Answer: thank you for your comments. Using high throughput approach, such as the chip array, it is normal that an important rate of marker is discarded. As reported in many papers (Wang et al., 2014 - https://doi.org/10.1111/pbi.12183; Wen et al., 2017 - https://doi.org/10.3389/fpls.2017.01389; Fiore et al., 2019 - https://doi.org/10.3390/plants8050116; Taranto et al., 2020 - https://doi: 10.3389/fgene.2020.00217; Mercati et al., 2021 - https://doi: 10.3389/fpls.2021.692661; Taranto et al., 2022 - https://doi.org/10.3390/agronomy12061326; …and more) a strong filtering is applied to select only the high informative SNPs to avoid the noise due to the low quality or low polymorphic markers. The filtering used is well described in the text in the “4.4 DNA Extraction and SNP Genotyping” section and the filtered SNP profiles for each sample analyzed in this manuscript is already included in the Supplementary material (Table S1).

- The abstract needs to be improved and should clearly indicate the results like how many sub-populations were detected in the whole panel and the small panels used and what inference was deduced.

Answer: thank you for your suggestions. We improved the abstract by reporting more clearly the results and the flow of the analyses performed.